# ADVERSARIAL LEARNED FAIR REPRESENTATIONS USING DAMPENING AND STACKING

## ABSTRACT

As more decisions in our daily life become automated, the need to have machine learning algorithms that make fair decisions increases. In fair representation learning we are tasked with finding a suitable representation of the data in which a sensitive variable is censored. Recent work aims to learn fair representations through adversarial learning. This paper builds upon this work by introducing a novel algorithm which uses dampening and stacking to learn adversarial fair representations. Results show that that our algorithm improves upon earlier work in both censoring and reconstruction.

## 1 INTRODUCTION

The need to have machine learning algorithms that make fair decisions becomes increasingly important in modern society. A decision is fair if it does not depend on a sensitive variable such as gender, race, or age. Models trained with biased data can lead to unfair decisions Mehrabi et al. (2021). In fair representation learning we are tasked with finding a suitable representation of the data in which the sensitive variable is censored. This ensures that these representations can be used for any downstream task, such as classification or segmentation, which should not rely on the value of the sensitive variable. Throughout this paper, we often refer to this sensitive variable as the *protected variable*.

Fairness can be applied to machine learning algorithms at roughly three stages of the process: during *preprocessing*, *inprocessing* or *postprocessing*. With preprocessing we aim to learn a new representation of the input data which is more fair. A well known example of this is Zemel et al. (2013), which obfuscates inputs when it can lead to unfairness. With inprocessing techniques the task is to make a machine learning algorithm more fair during training, typically by modifying the learning algorithm or by adding extra constraints to the learning objective. With postprocessing we are trying to correct the predictions of a machine learning algorithm after training in order to achieve fairness. In recent years, inprocessing techniques such as Zhang et al. (2018) have become very popular since they typically strike an optimal balance between accuracy and fairness. However, the major advantage of preprocessing over any other technique is that the transformed data can be used for any downstream task, both supervised and unsupervised. This makes preprocessing still invaluable in many practical applications where we know the protected variable beforehand, but have no specific machine learning task in mind yet. Hence the focus on preprocessing in this paper. Moreover, as shown in McNamara et al. (2017), preprocessing techniques can still provide us with theoretical fairness guarantees if required.

It is important to note that the notion of fairness is not trivial, and a multitude of fairness constraints have been proposed pertaining to both group fairness and individual fairness Mitchell et al. (2021). In this paper we adopt the *demographic parity* constraint due to its widespread use in benchmarking and evaluating fairness of machine learning algorithms. Demographic parity enforces that a classifier treats the data containing the protected variable statistically similar to the general population, and a major downside of this criterion is that it tends to cripple accuracy as long as we achieve equal acceptance rates. In reality, for every specific dataset and problem we need to assess which fairness criterion is most applicable and cannot simply select one as preferred Verma & Rubin (2018). The upside however is that in this paper we encode our fairness constraint in the form of a *loss function*, and as shown in Madras et al. (2018) we are able to associate different loss functions to different

group fairness constraints. This makes this approach applicable to far more fairness metrics than the one that we adopted in this paper.

Often with learning a fair representation, the naive approach of dropping certain features of the data is insufficient. The origin of the bias might latently depend on some nonlinear combination of other variables, and can thus leak back into a decision making model. This inspired the work by Edwards & Storkey (2016), which aims to learn a fair representation through adversarial learning. They use an auto-encoder as a generator for the new representation whose aim is to learn a new latent representation which attempts to censor the protected variable for the adversary. This work was later extended in Madras et al. (2018) where they propose learning objectives for other fairness metrics such as equalized odds and equal opportunity. In Kenfack et al. (2021) this work was further extended by introducing stacked auto-encoders to enforce fairness and improve censoring at different latent spaces.

This work builds on the previous adversarial approach. In particular it focuses on the case where the downstream task we may encounter is unknown, i.e. it can be either some supervised classification objective or some unsupervised clustering or segmentation objective. The challenge with learning fair representations is that on one hand we want to censor the data, and on the other we want to retain as much information as possible. Since these objectives are often opposed, the approaches in Edwards & Storkey (2016), Madras et al. (2018), Kenfack et al. (2021), and various others define the global objective of the model as a weighted sum of reconstruction error and predictive loss. This requires the trainer of a model to select some suitable hyperparameter which defines how much we value reconstruction error over predictive loss. This hyperparameter often has a large impact on the learned representations we get, and we can identify at least three issues with it. Firstly, we have no a priori knowledge on how the reconstruction error and the predictive loss relate. It could be nonlinear, which makes it almost impossible to make an informed decision beforehand. Secondly, the value of this hyperparameter gives us no formal guarantee of the censoring capabilities of the model. Some values can cause a collapse of the model. Thirdly, the hyperparameter choice is not explainable to the relevant stakeholders of the model. This makes it impractical for most industry use cases where hyperparameter choices need to be justified. As such, many authors using this methodology such as Edwards & Storkey (2016), Beutel et al. (2017), Madras et al. (2018), Feng et al. (2019), Kenfack et al. (2021) use a trial-and-error approach, or an arbitrary chosen constant, with regard to the choice of this hyperparameter. More often than not, the censoring capabilities of the learned representation are a hard constraint of the model. Thus, in many industry use cases, we are only interested in finding solutions in some restricted hypothesis space abiding some censoring constraint.

A second perhaps even greater issue with the previous work is its instability. In particular, due to the unstable dynamic between actor and adversary we often learn suboptimal solutions. This has been observed in many cases such as Edwards & Storkey (2016) and Kenfack et al. (2021), but never fully addressed. This paper attempts to mitigate these issues by introducing a novel algorithm for learning fair representations. In particular, it uses dampening to stabilize the interaction between actor and adversary, and uses stacking to learn strong censored representations within a restricted hypothesis space.

The remainder of the paper is structured as follows: in Section 2 we briefly reiterate related work, in Section 3 we formally define the problem, in Section 4 we introduce the algorithm, in Section 5 we discuss the experiments and results, and in Section 6 we conclude this work.

## 2 RELATED WORK

In Zemel et al. (2013) the first fair representation learning approach was presented. Their methodology aims to map input data to a new representation in terms of a probabilistic mapping to a set of prototypes. Several other noteworthy algorithms for finding fair representations are further explored in Feldman et al. (2015) and Calmon et al. (2017).

In Louizos et al. (2016) an architecture based on the Variational Auto-Encoder (VAE) was proposed in order to learn fair representations, called the Variational Fair Auto-Encoder. A similar idea is explored in Locatello et al. (2019). Although the idea of disentanglement between the protected variable and other features seems promising, it has not found widespread use yet due to the difficulty of finding independence between the sensitive and latent factors of variations.

In Edwards & Storkey (2016) the first adversarial approach was introduced to learning fair representations. They use an auto-encoder as a generator for the new representation whose aim is to learn a new latent representation which attempts to censor the protected variable for the adversary. Many paper in the fairness community have followed this line, noteworthy Beutel et al. (2017) and Feng et al. (2019). This work was extended by Madras et al. (2018) where they propose learning objectives for other fairness metrics. In Kenfack et al. (2021) this work was further extended by introducing stacked auto-encoders to enforce fairness at different latent spaces.

## 3 PROBLEM DEFINITION

This paper focuses purely on representation learning rather than classification. The aim is to learn a fair representation independent of the downstream that may be encountered (supervised or unsupervised). The notation of Edwards & Storkey (2016) of using the letter $X$ to represent the data, and $S$ to represent the protected variable is adopted. Each $x_i \in X$ is assumed to be some real-valued vector $x_i \in \mathbb{R}^n$, and each $s_i \in S$ is either 0 or 1, denoting if instance $i$ is sensitive or not: $s_i \in \{0, 1\}$. As argued in the introduction, the *demographic parity* constraint is adopted Dwork et al. (2012): given data $X$ and protected variable $S$, the aim is to learn a new representation $f(X)$ for which it holds that for any predictor $g$ derived from $f(X)$ we have $g(f(X)) \perp S$, i.e. $g(f(x))$ and $S$ are independent. In short, the aim is to find a representation $f(X)$ which give no predictive preference towards $S$. Throughout this paper $f(X)$ is referred to as the *censored* representation. It is important to note that the censored representation is not (necessarily) in the same space as the original data, and can have a different number of dimensions.

On one hand the aim is to censor the representation, while on the other hand the goal is to retain as much information as possible. In order to capture these opposing objectives, the learning objective can be framed as an adversarial learning problem. Similar to Edwards & Storkey (2016) and various papers following this, two agents are modeled with competing objectives: An auto-encoder $e$ with corresponding decoder $d$ representing the actor; and a classifier $h$ representing the adversary. As per usual, $e$, $d$ and $h$ are implemented in this paper using a feed-forward neural network. The aim is to find a censored representation $e(X)$. The objective of the adversary is to predict $S$ from the censored representation $e(X)$, while the aim of the actor is to learn this censored representation such that $d(e(X))$ is as close to $X$ as possible (the normal objective of an auto-encoder) *and* to deny the adversary from being able to learn $S$ from $e(X)$.

To make these notions precise, let $\mathcal{L}_{e,d}^{act}$ be the loss of the auto-encoder, and set it to be the mean-squared error (MSE), alternatively reconstruction error:

$$\mathcal{L}_{e,d}^{act} = \frac{1}{|X|} \sum_{x_i \in X} \|x_i - d(e(x_i))\|_2^2$$

Moreover, the loss of the adversary is set to be the *negative* cross-entropy loss over $S$:

$$\mathcal{L}_{e,h}^{adv} = \frac{1}{|X|} \sum_{s_i, \hat{s}_i \in S, h(e(X))} s_i \, log(\hat{s}_i) + (1 - s_i) \, log(1 - \hat{s}_i)$$

Note that in Madras et al. (2018) a multitude of other loss functions are proposed which lead to different notions of fairness (e.g. equal opportunity or equalized odds), but as argued earlier this paper optimizes for parity. Defining the loss to be the negative cross-entropy allows us frame it as a maximization problem rather than a minimization one, which entails that the joint objective of the actor and adversary is in the form of a min-max problem. Particularly, let $\mathcal{L}(e, d, h)$ be the joint loss, and define it as a weighted sum of $\mathcal{L}_{e,d}^{act}$ and $\mathcal{L}_{e,h}^{adv}$:

$$\mathcal{L}(e, d, h) = \mathcal{L}_{e,d}^{act} + \alpha \mathcal{L}_{e,h}^{adv}$$

Here $\alpha$ is some predetermined chosen hyperparameter denoting the importance of $\mathcal{L}_{e,d}^{act}$ over $\mathcal{L}_{e,h}^{adv}$. Since the negative cross-entropy loss is considered, the aim is to *minimize* this loss under the assumption that the adversary is trying to *maximize* this. Thus, the aim is to find $e$ and $d$ which satisfy the following:

$$\min_{e,d} \max_{h} \mathcal{L}(e, d, h)$$

Once these $e$ and $d$ have been found, the fair representations can be computed with $e(X)$ and the *censored original* representation by $d(e(X))$. Reason to use the latter can be due to the fact that these representations share a lot of the inherent properties of $X$ both dimension- and structure-wise. Thus, depending on the use case and task, $e(X)$ or $d(e(X))$ can be used to characterize the censored data.

## 3.1 Restricting the Hypothesis Space

A problem with the joint loss function $\mathcal{L}(e, d, h)$ is the correct choice of $\alpha$. This hyperparameter needs to be selected beforehand, and it has a large impact on the representations that are learned. Since (1) there is typically no a priori knowledge on how $\mathcal{L}^{act}_{e,d}$ and $\mathcal{L}^{adv}_{e,h}$ relate for a given $X$ and $S$, (2) choices of $\alpha$ give no formal guarantee on the censoring ability of the encoder, and (3) a choice of $\alpha$ is hard to explain to the relevant stakeholders, any choice of $\alpha$ is hard to justify and interpret. Additionally, low and high values for $\alpha$ could cause the trivial function to be learned, i.e. either the encoder learns an uncensored representation, or a constant function is learned.

In order to eliminate these problems, we propose a different objective function. More often than not, the censoring capabilities of the target function are a hard constraint on the resulting model. We recognize that perfect censoring is in most cases not feasible, and as such these hard constraints should define a hypothesis space of possible target functions. An example hard constraint could be that we do not wish that a very competent adversary receives above 60% accuracy on trying to classify the gender based on a loan application. Such a hard constraint solves the problem of not having a formal guarantee of the target function, and is both more intuitive and explainable to the relevant stakeholders.

To formalize this, a score function $score_{X,S,e}(h)$ and accompanying threshold $T$ is assumed to be specified beforehand which evaluates the performance of adversary $h$ based on data $X$, $S$ and encoder $e$. The constrained hypothesis space $\hat{E}$ for $e$ can be defined as follows:

$$\hat{E} = \{e \mid \underset{X,S,e}{score}(\arg\max_{h}(\mathcal{L}^{adv}_{e,h})) \leq T\}$$

It is assumed that $score$ and $T$ are chosen in accordance with the distribution of $S$ such that $\hat{E}$ is nonempty, e.g. if an accuracy score is used, $T$ should at least be 50%. The global objective is now to simply to minimize $\mathcal{L}^{act}_{e,d}$ by only considering encoders from the viable hypothesis space $\hat{E}$, that is to optimize $\min_{e \in \hat{E}, d} \mathcal{L}^{act}_{e,d}$. In order to find solutions in $\hat{E}$, different optimization methods need to be used. One of the main contributions of this paper is that such an algorithm is supplied.

## 4 Adversarial learning using Stacking and Dampening

Before delving into the technical details, it is worthwhile to discuss the shortcomings of the current approach. As mentioned in Edwards & Storkey (2016), Madras et al. (2018), and various other papers, it is very difficult to train these models due to the unstable dynamic between the actor and the adversary. This is true for adversarial learning in general because of the underlying saddle point optimization problem. Different approaches in literature have been proposed to stabilize adversarial networks, ranging from simple solutions such as early stopping or weight clipping Arjovsky et al. (2017) to more intricate ones such as adding extra stabilizing steps Yadav et al. (2018). A multitude of stabilization methods exist, but the downside is that they need to be adapted accordingly to the properties of the loss function Xing et al. (2021). Thus, in the context of our setting, it is worthwhile to investigate what the root cause of the instability is. During training, the actor is continuously updating in the direction to make the adversary worse at predicting the protected variable (recall that the objective of the actor is to minimize the *negative* cross-entropy loss of the adversary, while the adversary is trying to *maximize* this). A key insight here is that the loss signal that the adversary is giving to the actor is paradoxical in nature:

- If the magnitude of the loss is high, the adversary is incompetent at predicting $S$. Since the adversary is also updating its own loss towards 0, it means we are at a point where the gradient of the loss will be high. This in turn will result in a big update of the actor.

However, the adversary was already incompetent at predicting $S$, but a big update in an uninformative direction is performed as opposed to a smaller conservative update.

- If the magnitude of the loss is relatively low, the adversary is competent at predicting $S$. When the loss is relatively smooth at local maxima, a low magnitude of the loss will more often than not result in a small update of the actor. However, the adversary was already competent at predicting $S$, but we are performing a small update in an informative direction when we would rather perform a bigger less careful update.

The key issue is thus that if the adversary is too competent then the gradients will be weak, and if the adversary is too incompetent the gradients will be uninformative. This interplay between the competence of the adversary and the size of the gradients is also mentioned in Edwards & Storkey (2016), but not further explored.

Now consider what this means when we are actually training and updating the actor and adversary. When the weights of the actor and adversary are adjusted in turns, as described in Goodfellow et al. (2014), there is a risk that the adversary will never be sufficiently competent in the task. This is particularly true when a strong adversary is used with a lot of parameters, which typically need more batches to converge. This means that constantly big weight adjustments in an imprecise direction are made, which again causes the adversary to be incompetent. On the other hand when the adversary is trained in the inner loop, apart from it being very inefficient, we would also run the risk of the adversary being too strong, and not being able to make any meaningful updates.

To mitigate these problems and to make the training process more stable, the notion of *dampening* is introduced.

## 4.1 DAMPENING

Dampening is a function that serves as a modulating term of our algorithm in the interaction between actor and adversary. Similar ideas for weighted loss schemes such as focal loss have been proposed for imbalanced datasets Lin et al. (2017), but here it is applied to modulate the actor and adversary. Dampening returns a number between 0 and 1 denoting how much information the classifier has over a training sample. First, define $g$ as a function over subsets $S'$ of our protected variable $S' \subseteq S$:

$$g(S') = \frac{1}{|S'|} \max \left( \sum_{s_i' \in S'} s_i', \sum_{s_i' \in S'} 1 - s_i' \right)$$

In words, $g(S')$ represents the best possible accuracy a predictor can receive when using only information about $S'$. Observe that since the protected variable is binary, $g(S') \in [0.5, 1]$. The role of $g(S')$ is to serve as a baseline guessing accuracy.

Now given a classifier $f$ and training sample $X', S'$, dampening $\delta$ is defined as:

$$\delta(f, X', S') = \frac{\max(0, acc(f, X', S') - g(S'))}{1 - g(S')}$$

In the above definition, $acc(f, X', S')$ is used as shorthand notation to denote the accuracy score of $f$ on training sample $X', S'$. Whenever $g(S') = 1$, we set $\delta(f, X', S') = 0$. In words, dampening $\delta(f, X', S') \in [0, 1]$ tells us the percentage decrease of number of misclassifications would we use $f$ instead of guessing the most frequent label in the sample. Whenever dampening is 1 for $f$, we know that $f$ achieves perfect accuracy on the training sample $X', S'$, and whenever dampening is 0 we would be no worse off by just informed guessing. Thus, dampening is a measure of information a classifier has over a certain classification task. An important property of dampening is that it is contained within the unit interval, meaning that it can be used as a modulating term since the results will never be larger than the original value. Multiple notions of bounded information were explored, but found dampening to work the best for a variety of tasks. We suspect it is due to its linear scaling with the number of correctly classified samples whenever its value is nonzero; a small increase in correctly classified samples translates in a small increase in dampening, and vice versa for a big increase.

---

**Algorithm 1** ALFR-DS

---

Initialize $e = id$, initialize $\theta^{act}$ and $\theta^{adv}$ randomly.      ▷ Start with the "empty" encoder.
**repeat**
    Initialize $e_{new}$ randomly.
    $e = e_{new} \circ e$      ▷ Add new encoder to the (frozen) stack.
    **repeat**
        $X', S' = $ random mini-batch from $X, S$
        $L^{act} = \mathcal{L}_{e,d}^{act}(X')$
        $L^{adv} = \mathcal{L}_{e,h}^{adv}(X', S')$
        $\theta^{act} = \theta^{act} - \eta \cdot \left( \nabla_{\theta^{act}} L^{act} + \delta(h \circ e, X', S') \cdot \nabla_{\theta^{act}} L^{adv} \right)$
        $\theta^{adv} = \theta^{adv} + \eta \cdot \left( 1 - \delta(h \circ e, X', S') \right) \cdot \nabla_{\theta^{adv}} L^{adv}$
    **until** Sufficient epochs reached.
    Freeze encoder $e$.
**until** $score_{X,S,e}(h) \leq T$ or Deadline reached.

---

## 4.2 STACKING

Stacking is a technique for censoring which was recently introduced in Kenfack et al. (2021). The idea is that during training we start out with a simple encoder which learns a censored representation. After this initial training phase, we freeze the encoder and append a new trainable one. This process continues until we are completely done with training. Another perspective on this process is that once we learn a censored representation, we recursively start over a completely new training process, except that we use the censored representation as the new input. The key idea behind stacking is that once a censored representation is learned and frozen, it is highly likely that *some* information about the protected variable is lost for good. Thus in theory, repeating the stacking operation can give us representation with arbitrary strong censoring properties.

It is important to note that the authors found that stacking did increase censoring over the original approach, but unsurprisingly also comes at the cost of reconstruction error. In other words, stacking should preferably be combined with a very careful and stable censoring algorithm, which in our case is handled by the addition of dampening. Stacking together with dampening serves the basis for our algorithm.

## 4.3 ALFR-DS

In Algorithm 1 a new algorithm is proposed called ALFR-DS ("Adversarial Learned Fair Representations using Dampening and Stacking"). This algorithm differs from basic ALFR, as discussed in Edwards & Storkey (2016), on three key aspects:

- A different loss function for the actor and adversary is used, instead of the same function $\mathcal{L}(e, d, h)$ as given in Section 3 for both.
- An inner loop for normal backpropagation is used, and an extra outer loop which incorporates stacking is added. An extra termination condition is added which allows us to find solutions in the restricted hypothesis space $\bar{E}$, as defined in Section 3.1.
- The actor and adversary are trained *concurrently* instead of interleaved.

In the description of the algorithm, $\mathcal{L}_{e,d}^{act}(X')$ and $\mathcal{L}_{e,h}^{adv}(X', S')$ is used to denote the loss functions defined in Section 3 applied to $X'$ and $S'$. Moreover $\theta^{act}$ and $\theta^{adv}$ refer to the model parameters of the actor ($e$ and $d$) and subsequently the adversary ($h$). The fixed learning rate $\eta$ can be replaced with a parameter-dependent dynamic one: in all of the experiments the Adam optimizer has been found to work the best Kingma & Ba (2014).

The role of dampening in the algorithm is to act like a "fuzzy" turn-taking mechanism: whenever the adversary is weak, $\delta$ will be close to $0$ in our algorithm. This means the actor will hardly use the loss of the adversary in updating the censored representations, i.e. it will act like a normal auto-encoder. Since representation learning in a normal auto-encoder is stable, it gives the adversary time to learn and catch up. Whenever the adversary is strong, $\delta$ will be close to $1$, meaning the adversary will hardly update itself. This allows the auto-encoder to incorporate the loss of the adversary and

learn a new censored representation. In other words, the actor can catch up. This is how dampening attempts to stabilize the learning process: it gives either the actor or the adversary time to catch up, resulting in only informative updates of the model. In the experiments no extra stabilizing methods such as gradient clipping were needed.

The width and depth of the encoder that is added to the stack can be varied at any moment. In our implementation, every subsequent encoder after the first uses the same input and output size. In order to be able to censor nonlinear relations in the data at every step, every encoder is given a single hidden layer. It is important to note that the adversary $h$ can have any neural architecture, depending on the desired censoring strength of the resulting model. It is also important to note that a strong adversary typically means a longer training period for convergence.

The termination condition $score_{X,S,e}(h) \leq T$ tells us when an encoder is contained within the desired hypothesis space given adversary $h$. Although not explicitly mentioned in the algorithm, it is sometimes beneficial to fully train $h$ on the training data $X, S$ without additionally training the encoder after termination of the inner loop. This is to ensure that the adversary is fully converged before an assessment about the censoring capabilities of the model is made. Under the reasonable assumption that $score$ is chosen in such a way that it eventually decreases as $\mathcal{L}_{e,h}^{adv}$ decreases, and that $\mathcal{L}_{e,h}^{adv}$ decreases after a completion of the inner loop, it can be concluded that $score_{X,S,e}(h) \leq T$ will eventually hold. In other words, ALFR-DS will eventually find a solution in the constrained hypothesis space. In order to encourage that $\mathcal{L}_{e,h}^{adv}$ decreases after a completion of the inner loop, the hidden size of each encoder that is added to the stack can be decreased, or the inner loop can be terminated early. However, since it is often undesirable that the model complexity of the encoder grows arbitrarily large, an extra deadline criterion is used to allow early stopping. Whenever an early termination occurs, a solution with the desired censoring capabilities was not found within the complexity bounds of the model.

## 5 EXPERIMENTS AND RESULTS

This section consists of two different experiments that were conducted. Motivated by the image anonymization task proposed in Edwards & Storkey (2016), the aim of the first experiment is to closely compare ALFR with ALFR-DS on a widely used image dataset (MNIST; Deng (2012)). The second experiment is more general and aims to compare several variants of ALFR, ALFR-DS and other preprocessing algorithms from the widely used IBM AI Fairness 360 package Bellamy et al. (2018) on several widely used fairness-related datasets.

### 5.1 MNIST

The dataset contains 60,000 handwritten images with corresponding labels. The goal is to censor all the 8s in the dataset, i.e. the protected variable $s_i$ is set to 1 whenever the label is 8, and 0 otherwise. Even though this task does not serve a direct practical use, it is a great benchmark for its inherent challenging properties. In particular, the protected variable is *unevenly distributed* and the task is very *nonlinear* in nature. Moreover, due to its wide spread it allows for easy replication.

In this experiment the goal is to compare ALFR-DS to ALFR in its ability to reconstruct and censor. In order to ensure a fair comparison, every model was trained for 30 epochs. For ALFR-DS, 3 variants were considered: **ALFR-DS(1)** which runs the inner loop of Algorithm 1 once for 30 epochs, **ALFR-DS(2)** which runs the inner loop twice for 15 epochs, and **ALFR-DS(3)** which runs the inner loop three times for 10 epochs. Both ALFR-DS and ALFR used the same Multi-Layer Perceptron (MLP) adversary. However, we observed that ALFR typically performs better against a weak adversary, so we also considered a variant of ALFR against an adversary using simple Logistic Regression (LR). These two variants are referred to as **ALFR (MLP)** and **ALFR (LR)**. For ALFR several values for $\alpha$ were tried, but only report for $\alpha = 1$ since ALFR-DS outperforms ALFR for all nontrivial choices of $\alpha$ in both censoring and reconstruction. Both ALFR-DS and ALFR were given the same number of target dimensions to embed to. In order to measure how well a model censors, each model was trained on one slice of the data, after which another slice was used for evaluation. Two new classifiers, one using **LR** and one using an **MLP**, were freshly trained on these new representations and were asked to predict protected variable $S$. The resulting accuracy scores were used as a benchmark (lower is better). Additionally, a normal auto-encoder was trained that did

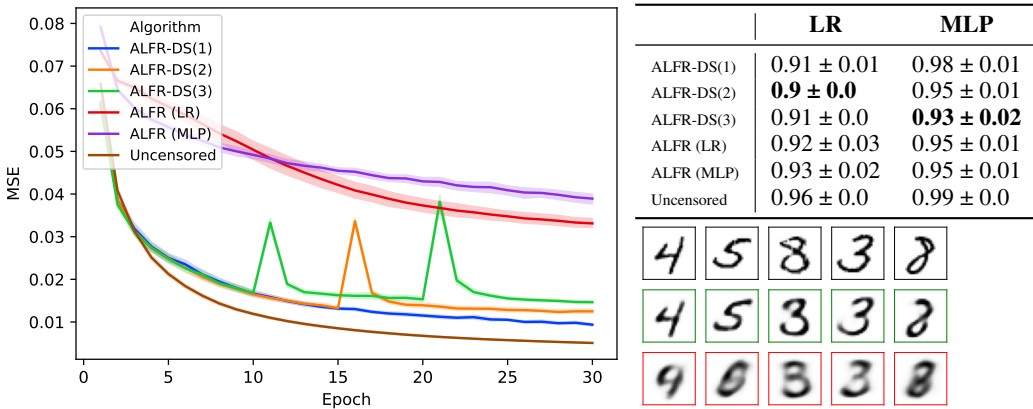

Figure 1: Left shows the reconstruction error (MSE) over 30 epochs for the MNIST task. Top-right lists all the accuracy scores of different adversaries with increasing strength trained on the final representation, where lower is better. Bottom-right shows some examples of the original MNIST digits (black outline at the top) which were both censored by ALFR-DS(3) (green outline in the middle) or by ALFR (LR) (red outline at the bottom).

not use an adversary and called this **Uncensored** (which is equivalent of using ALFR with $\alpha = 0$). All experiments were repeated 10 times to account for naturally occurring deviations in the results.

In Figure 1 the result of the experiment can be seen, which also provides a visual output of ALFR-DS(3) versus ALFR (LR). In one instance ALFR-DS censors an 8 to a 3 and in another instance an 8 to a 6. From this visual inspection, it is clear that a lot more information is lost about the original images with ALFR. Moreover, an "imprint" of an 8 across most images in the censored representations can be observed. The "spikes" in the reconstruction graph occur at the times in which a new encoder is added to the stack; in this case the loss briefly rises since each new encoder is initiated randomly. It is also interesting to note is that for the first 3 to 5 epochs, the graphs of ALFR-DS and Uncensored are almost identical due to the fact that especially in the early phases dampening will be relatively small. The learned representations are still evolving a lot, meaning that the adversary has no time to become competent at predicting $S$. Only when the representations start to stabilize after approximately 5 epochs, a difference in the graphs can be observed.

The lowest theoretical accuracy that can be achieved due to the distribution of $S$ is 0.9 for this task and the highest possible accuracies are reported for Uncensored. Again ALFR-DS achieves superior results in both censoring and reconstructing over ALFR. It is clear from the results that the number of stacked encoders in ALFR-DS does come at the cost of reconstruction. Particularly, ALFR-DS(1) is outperformed by ALFR-DS(3) in terms of censoring, however ALFR-DS(1) leaves much more of the data intact. It is thus clear that a balance has to be struck between censoring and reconstruction, which is normally handled by the algorithm with the use of the *score* function with accompanying threshold $T$. In both tasks when it comes to censoring, ALFR-DS(2) and ALFR-DS(3) perform similarly except when a strong nonlinear MLP adversary was tasked to predict $S$, in which case it becomes clear that ALFR-DS(3) has greater censoring capabilities. It is noteworthy to observe that ALFR(LR) outperforms ALFR(MLP) for censoring even though a weaker adversary was used during training. The reason is that LR converges much faster, and thus gives a more informative loss signal to the actor after each turn. Due to dampening, ALFR-DS does not have this problem and can reliably be used against strong adversaries.

## 5.2 AIF360 BENCHMARK

To really see how ALFR-DS performs in practice, it is compared to several preprocessing algorithms on several datasets using AIF360. Finding the right comparison is non-trivial, since ALFR-DS is designed to learn a censored representation for *any* target variable as opposed to one specific variable. In practice this means that ALFR-DS does not have access to the target variable at training time, putting it at a major disadvantage. Taking this into consideration, it is still worthwhile to investigate how ALFR-DS compares.

| Dataset | **Adult** | | | | **German** | | | | **Bank** | | **COMPAS** | | | |
|---|---|---|---|---|---|---|---|---|---|---|---|---|---|---|
| Protected | **Sex** | | **Race** | | **Sex** | | **Age** | | **Age** | | **Sex** | | **Race** | |
| Metric | BA | $\Delta_{DP}$ | BA | $\Delta_{DP}$ | BA | $\Delta_{DP}$ | BA | $\Delta_{DP}$ | BA | $\Delta_{DP}$ | BA | $\Delta_{DP}$ | BA | $\Delta_{DP}$ |
| ALFR-DS(1) | 69 | 17 | 68 | 8 | **63** | **5** | **62** | **10** | **60** | **3** | 66 | 14 | 64 | 15 |
| ALFR-DS(2) | 68 | 15 | 66 | 8 | *58 | *4 | *58 | *8 | *58 | *3 | 65 | 13 | 64 | 13 |
| ALFR-DS(3) | **67** | **14** | 67 | 6 | *51 | *2 | *57 | *7 | *57 | *3 | **63** | **11** | 63 | 12 |
| ALFR-S(1) | 73 | 18 | 73 | 9 | *55 | *5 | *57 | *6 | *57 | *3 | 66 | 12 | 65 | 16 |
| ALFR-S(2) | 70 | 17 | 68 | 8 | *55 | *5 | *53 | *7 | *51 | *1 | 61 | 11 | 62 | 12 |
| ALFR-S(3) | 69 | 17 | 69 | 8 | *55 | *5 | *54 | *7 | *51 | *1 | *58 | *8 | *58 | *11 |
| DIR | 75 | 17 | 76 | 9 | 65 | 6 | 68 | 11 | 83 | 5 | 67 | 15 | **67** | **8** |
| LFR | *50 | *0 | *50 | *0 | *50 | *0 | *50 | *0 | *51 | *2 | *50 | *1 | *50 | *1 |
| Uncensored | 78 | 18 | 78 | 9 | 66 | 7 | 70 | 16 | 75 | 8 | 68 | 20 | 68 | 18 |

Table 1: Results (%). Best are given in bold and discarded ($< 60\%$ BA) marked with *.

This experiment is conducted on several datasets and protected variables. For each combination, a censored representation is learned on 80 percent of the data using the algorithm of choice. Afterwards a gradient boosting classifier was trained on these representations to predict the target variable. The performance was evaluated on the remaining 20 percent. Due to imbalanced datasets, the balanced accuracy (BA) was reported together with the demographic parity distance ($\Delta_{DP}$), which is the proportional distance of positive outcomes between the privileged and unprivileged groups. To strike a balance between censoring and accuracy, only results with a BA of $60\%$ are considered. This is to remove all cases where there is too much information removed (e.g. the trivial uni-label function). From these candidates the one with the lowest $\Delta_{BP}$ is elected as the winner.

To analyze the individual effects of stacking and dampening, an ablation study is performed where **ALFR-S(n)** is default ALFR performed with $n$ stacks, and **ALFR-DS(n)** ALFR-DS with $n$ stacks. Thus, vanilla ALFR is equal to **ALFR-S(1)** and ALFR-DS without stacking is equal to **ALFR-DS(1)**. Additionally, Learning Fair Representations (**LFR**) Zemel et al. (2013), and Disparate Impact Remover (**DIR**) Feldman et al. (2015) are compared. Note that the removal of disparate impact is equivalent to achieving demographic parity. The optimized stochastic preprocessing technique from Calmon et al. (2017) was not benchmarked due to unavailability of the required preprocessing and distortion functions for all datasets. The datasets that were used are UCI Adult (predict the income of a person) Asunción & Newman (2007), German (predict defaults on consumer loans), Bank Marketing (predict subscription of a term deposit) Moro et al. (2014), and the well-known COMPAS dataset (predict recidivism).

In Table 1 the average percentage results of 10 runs are shown. In cases where careful censoring is needed to stay above 60% BA, ALFR-DS(1) gives good results (German and Bank). In cases where there is more room for censoring, ALFR-DS(3) performs better. Whenever accuracy outweighs fairness, DIR tends to outperform ALFR-DS, but note that DIR is conditioned on the target variable and in some cases (Adult) stays close to the uncensored representations. Only in the case of the COMPAS + race task does DIR outperform ALFR-DS in terms of censoring. ALFR-S(n) either seems to censor too strongly (German, Bank), or not strong enough (Compas, Bank). Moreover, LFR seems to give poor results overall, even after numerous attempts to optimize the hyperparameters. Overall, ALFR-DS(n) gives the best results when a careful balance between censoring and accuracy is needed, where higher $n$ implies a stronger censoring.

## 6 CONCLUSIONS

In this paper, we have given a novel algorithm that uses dampening to stabilize the interaction between actor and adversary, and uses stacking to learn strong censored representations within a restricted hypothesis space. This algorithm outperforms the current approach in both censoring and reconstruction, as shown in our empirical results.

Since the aim is to learn meaningful representations which can be learned for fair downstream tasks, we believe that a good comparison should be made with VAE's that attempt to learn these disentangled representation. Moreover, due to the empirical nature of this study, theoretical bounds on reconstruction and censoring using stacking and dampening should be further explored.

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
