# OpenReview forum: "Adversarial Learned Fair Representations using Dampening and Stacking"
_ICLR.cc/2023/Conference — Submitted to ICLR 2023_

### Official Review · Reviewer_kwp8 · 2022-10-24

**Confidence:** 4
**Correctness:** 2
**Technical Novelty And Significance:** 2
**Empirical Novelty And Significance:** 1
**Recommendation:** 3

**Clarity, Quality, Novelty And Reproducibility:**

The clarity and originality of this paper are fair, but the quality is relatively low. Please see my comments in Strength And Weaknesses.

**Strength And Weaknesses:**

Pros:

1. The paper considers an important problem of how to learn fair representations without knowing downstream tasks.

2. The paper is well-structured.

Cons:

1. The novelty of this work is limited. Compared with ALFR, the main novelty brought by ALFR-DS is the use of dampening together with some other tricks discussed in Section 4.3 to tame the instability of adversarial training. Given that ALFR was proposed 7 years ago and a variety of studies has explored similar issues in a more principled way [1, 2, 3], the contributions made in terms of methodology are inadequate.

2. The proposed dampening mechanism needs more support. In its current version, the paper does not provide sufficient theoretical grounding nor empirical analysis to verify that dampening does prevent instability. The only effort made is plotting the curve of the reconstruction error (Figure 1), which seems too coarse-grained to prove the effectiveness of dampening.

3. The writing is not clear enough. In Section 3 the problem formulation is entangled with the methodology. It is also hard to find a proper mathematical definition of “good” fair representations.

4. I have some questions regarding important details that are not well specified. For example, it is unclear how to select threshold $T$ and what value is used exactly for experiments. It also seems inexplicit why to exclude results with BA less than 60% for the AIF360 Benchmark.

[1] Roth, Kevin, et al. "Stabilizing training of generative adversarial networks through regularization." Advances in neural information processing systems 30 (2017).

[2] Yadav, Abhay, et al. "Stabilizing Adversarial Nets with Prediction Methods." International Conference on Learning Representations. 2018.

[3] Jenni, Simon, and Paolo Favaro. "On stabilizing generative adversarial training with noise." Proceedings of the IEEE/CVF Conference on Computer Vision and Pattern Recognition. 2019.

**Summary Of The Paper:**

The paper considers the problem of learning fair representations containing little or no information about the protected variable. In the absence of prior knowledge about downstream tasks, it requires the learned representations to be both fair and discriminative enough, which turns out to be a trade-off and may incur instability when applying adversarial training.

The main contribution of this paper is to propose a new algorithm called ALFR-DS that retrofits the existing adversarial learning framework ALFR [1] with dampening and stacking. Specifically, the algorithm uses an adaptive parameter (dampening) to stabilize the gradients of the actor and the adversary when the adversary is too weak and combines it with stacked encoders (stacking) [2] to improve censoring. Experimental results on MNIST and the AIF360 benchmark show that ALFR-DS achieves better censoring and reconstruction performance than several baselines.

[1] Edwards, Harrison, and Amos Storkey. "Censoring representations with an adversary." arXiv preprint arXiv:1511.05897 (2015).

[2] Kenfack, Patrik Joslin, et al. "Adversarial Stacked Auto-Encoders for Fair Representation Learning." arXiv preprint arXiv:2107.12826 (2021).

**Summary Of The Review:**

Overall, I think this paper does not present significant contributions and would recommend rejection. Please see my comments in Strength And Weaknesses.

---

### Official Review · Reviewer_4DEf · 2022-10-25

**Confidence:** 3
**Correctness:** 3
**Technical Novelty And Significance:** 2
**Empirical Novelty And Significance:** 1
**Recommendation:** 3

**Clarity, Quality, Novelty And Reproducibility:**

The clarity of the work needs some improvement. The work is, at a high level, reproducible from the description, though code would need to be released to achieve the same results quoted in the paper.

**Details Of Ethics Concerns:**

None noted.

**Strength And Weaknesses:**

Strengths:

The paper identifies and discusses a very interesting problem (fair representation learning). The background discussion in the introduction is easy to read and follow, though it may be more appropriate for a background or related works section. The idea of training until a fairness score T is met makes sense and is an interesting stopping criteria.

Weaknesses:

I start with the weaknesses in the presentation, then discuss some weaknesses in the methodology, then finally discuss weaknesses in the experimental evaluation.

The paper is not very well written, in my opinion. I will go into slightly more detail on methodological we  The abstract is sparse giving the reader little idea what exactly is explored and what the results are. The introduction is primarily a context for the work wit h little description of the aim of the work until the fifth paragraph of the introduction. The bullet point text in Section 4 is a bit convoluted and each time I read the paper I had to double take at what exactly it is trying to say. A minor stylistic note (but one that should be addressed) is that all of the citations here use \citet or \cite when they almost always should use \citep.

I think some intuition should be provided for the score function and T. Given that these values would be set by regulators or perhaps non-technical stakeholders I think further discussion of these values is important to give the reader an idea of exactly how these can be used in practice.

The experimental section is one of the biggest weaknesses of the paper. It is unclear the MNIST scenario studied fits into the problem framework set out by the authors. Are images of eights given a sensitive attribute vector that is all ones ($s_i$)? The results on the fairness benchmarks are much more intuitive, but the performance leaves something to be desired. Indeed the method improves fairness, but at a large cost to accuracy and it seems the authors have avoided comparing to the more recent benchmarks that are cited (Edwards & Storkey (2016), Madras et al. (2018)) instead opting for older and perhaps more simple baselines (unless I have missed something in this regard).






**Summary Of The Paper:**

In this paper the authors study the problem of learning a fair representation given a dataset. In order to learn fair representations, previous works have balanced the reconstruction loss with an adversarial loss using a weighting term. This weighting term is a central problem identified by the authors as blind selection of this weighting term can easily lead to trivial classifier performance (or unstable performance) and even if this weighting term is selected appropriately it can be difficult to justify to stake-holders. To remedy this, the authors propose to use dampening and stacking in order to develop stable fair representations with an informed choice of the weighting term. The authors then go on to experimentally evaluate this approach on MNIST and on standard fairness benchmarks.

Ultimately, the paper's writing is unclear in several places. The empirical results are not very strong, and it seems that they could have evaluated against stronger baseline methods.

**Summary Of The Review:**

While the paper aims at tackling an important problem (learning fair representations) its clarity and empirical evaluation need improvement for the impact of this work to be realized.

---

### Official Review · Reviewer_Hobp · 2022-10-29

**Confidence:** 2
**Correctness:** 3
**Technical Novelty And Significance:** 3
**Empirical Novelty And Significance:** 2
**Recommendation:** 5

**Clarity, Quality, Novelty And Reproducibility:**

* The paper introduces a modified loss-objective (Dampening) and couples it with existing approach of Stacking [Kenfack et al., 2021] in order to obtain better demographic parity based on preprocessing.
* Authors show using empirical evaluation that their approach performs better than existing approaches
* Theoretical fairness guarantee is not investigated for dampening-based modified loss, though as the authors write "preprocessing techniques can still provide us with theoretical fairness guarantees if required."


**Strength And Weaknesses:**

+ Clearly written paper for the most part.
+ MNIST experiments shows the influence of the approach with protecting censored data.
+ Mixed results on AIF360 dataset per my limited understanding - I do not follow the argument on ablation study and improvement obtained on dampening.

- Some recommendations: AIF360 benchmark and the dataset should be described better including what are the features, what is being censored. I strongly recommend adding a section to appendix/supplementary material to capture this information making the paper self-contained when reading the evaluation.


**Summary Of The Paper:**

The paper focusses on adversarial fair representation where the model should not rely on the sensitive variable based on a preprocessing approach. The authors highlight the problem with adversarial min-max loss leading to either  (a) high gradient and high loss in case of incompetent adversary; or (b) small loss and slow actor updates in the case of good adversary. They  introduce dampening, a new measure of information captured from a protected variable scaled to [0, 1] which balances between the min-max adverasarial loss objective between actor and adversary. This is coupled with stacking [Kenfact et al., 2021] where exist encoder is frozen (leading to loss of some information about the protected variable) and a new trainable single-layer encoder is appended to the existing encoder per training stage. Empirical evaluation on MNIST and AIF360 datasets shows censoring of protected variable (e.g., "8" in the MNIST dataset) using the ALFR-DS approach with the stacking helping (ALFR-DS(3) vs ALFR-DS(1)).

**Summary Of The Review:**

The paper presents a new loss objective (dampening) coupled with stacking as a preprocessing method and show better censoring as part of empirical evaluation. There is no theoretical fairness guarantee accompanying the new loss and the empirical evaluation section could be written better to aid understanding of how dampening helps.

---

### Decision · Program_Chairs · 2023-01-20

**Decision:**

Reject

**Justification For Why Not Higher Score:**

The proposed algorithm was not extremely novel or theoretically justified and the experimental support for the algorithm was not overwhelming.

**Justification For Why Not Lower Score:**

N/A

**Metareview: Summary, Strengths And Weaknesses:**

This paper is on the topic of learning fair representations with little information about the protected variable. There is a tradeoff between learning representations that are both fair and lead to high performance. One also seeks stability during adversarial training.

The main contribution of this paper is a new algorithm called ALFR-DS that extends the existing ALFR framework by adding dampening and stacking. In particular, an adaptive dampening parameter is introduced to stabilize the gradients of the actor and the adversary, and stacked encoders are used to improve censoring. Experiments on MNIST + AIF360 achieve improve censoring and reconstruction performance over baselines.

The main strength of the paper is that it addresses an important problem, namely learning fair representations without knowing the downstream tasks. Note that all the reviewers found the paper interesting and the direction certainly has potential for interesting results.

However, the main problems with this paper are:
* The algorithm was not very novel compared to ALFR.
* There was little theoretical motivation.
* The experimental results were not very compelling.
* There were issues with the presentation that were not addressed in the rebuttal.